# An Integrated Framework for Remote Sensing Assessment of the Trophic State of Large Lakes

**Dinghua Meng [1], Jingqiao Mao [1,\*], Weifeng Li [2], Shijie Zhu [3] and Huan Gao [1]**

[1] College of Water Conservancy and Hydropower Engineering, Hohai University, Nanjing 210098, China; dhmeng@hhu.edu.cn (D.M.); huangao@hhu.edu.cn (H.G.)

[2] State Key Lab of Urban and Regional Ecology, Research Center for Eco-Environmental Sciences, Academy of Sciences, No.18 Suangqing Rd., Beijing 100085, China; li.wf@rcees.ac.cn

[3] Department of Civil Engineering, The University of Hong Kong, Hongkong, China; sjzhu@connect.hku.hk

\* Correspondence: maojq@hhu.edu.cn

**Abstract:** The trophic state is an important factor reflecting the health state of lake ecosystems. To accurately assess the trophic state of large lakes, an integrated framework was developed by combining remote sensing data, field monitoring data, machine learning algorithms, and optimization algorithms. First, key meteorological and environmental factors from in situ monitoring were combined with remotely sensed reflectance data and statistical analysis was used to determine the main factors influencing the trophic state. Second, a trophic state index (TSI) inversion model was constructed using a machine learning algorithm, and this was then optimized using the sparrow search algorithm (SSA) based on a backpropagation neural network (BP-NN) to establish an SSA-BP-NN model. Third, a typical lake in China (Hongze Lake) was chosen as the case study. The application results show that, when the key environmental factors (pH, temperature, average wind speed, and sediment content) and the band combination data from Sentinel-2/MSI were used as input variables, the performance of the model was improved ($R^2$ = 0.936, RMSE = 1.133, MAPE = 1.660%, MAD = 0.604). Compared with the performance prior to optimization ($R^2$ = 0.834, RMSE = 1.790, MAPE = 2.679%, MAD = 1.030), the accuracy of the model was improved by 12.2%. It is worth noting that this framework could accurately identify water bodies in different trophic states. Finally, based on this framework, we mapped the spatial distribution of TSI in Hongze Lake in different seasons from 2019 to 2020 and analyzed its variation characteristics. The framework can combine regional special feature factors influenced by a complex environment with S-2/MSI data to achieve an assessment accuracy of over 90% for TSI in sensitive waters and has strong applicability and robustness.

**Keywords:** integrated framework; trophic state index; remote sensing; environmental factors; large lake; optimized machine learning algorithm

## 1. Introduction

Lake ecosystems are dynamic, nonequilibrium, nonlinear, and highly complex ecosystems which are particularly sensitive to human activities and climate change [1]. As large amounts of nitrogen and phosphorus nutrients resulting from chemical production, farmland irrigation, and rainfall enter lakes, the eutrophication level of the water rises, and harmful algae and invasive species of organisms multiply rapidly, eventually leading to outbreaks of algal blooms [2]. Therefore, the trophic state is a crucial issue influencing the water environment of lakes. According to a study of the trophic state of 84 typical large lakes in China, more than 85% of the lakes were in the eutrophic state, and the remainder were in the mesotrophic state [3,4].

Water quality monitoring is essential to ensure the safety of water resources and prevent water pollution. Traditional field monitoring [5,6] has the benefits of high monitoring accuracy and dependable data; unfortunately, it also has disadvantages such as

being time-consuming, expensive, and having a limited testing range. Therefore, the long-term dynamic monitoring of water quality indicators is limited by constraints [7,8], and rapid identification of pollution sources is difficult [9]. Since the 1970s [10], remote sensing technology has been gradually applied to the dynamic monitoring of water quality because of its efficiency, wide coverage, and low cost. Therefore, effectively combining these approaches can compensate for the limitations of field monitoring, making it possible to dynamically monitor lake water quality indicators on a large scale and in long time series [11,12]. In addition, it can efficiently identify pollution sources, which has a positive effect on the economic and social development of lake basins [13].

To apply this technique to the trophic state of lakes, existing studies have generally developed statistical regression models based on remote sensing and field monitoring data [14]. Then, certain water quality indicators such as chlorophyll-a (Chl-a) [15,16], dissolved oxygen (DO) [17], and $NH_3$-N [18] are inverted to assess the distribution of the trophic state of lakes. Although the inversion accuracy of this method is high, the use of a single indicator [19] as the standard for assessing the trophic state requires additional validation because the lake is influenced by multiple factors. The use of remote sensing to invert Chl-a in lakes has become a popular research topic [18]. However, Chl-a is more appropriate for early warning of harmful algal blooms [20]. In the absence of substantial deterioration of the environment of lakes, Chl-a may have limited application. Furthermore, most of existing studies estimated the trophic state of lakes by assigning or calculating the weights of multiple indicators [21], which results in an accumulation of errors during the inversion, thereby increasing the uncertainty of the assessment results [15].

To reduce errors in the assessment of the trophic state of lakes, researchers have proposed the trophic state index (TSI) [22], which is gradually being applied in trophic state analysis and for risk prediction of algal bloom outbreaks. This index considers five indicators: Chl-a, total phosphorus (TP), total nitrogen (TN), Secchi depth (SD), and the permanganate index ($COD_{Mn}$) [23]. The TSI can be quickly calculated to identify and compare the trophic state of lakes according to grading criteria. Unfortunately, because the TSI involves several indicators reflecting the trophic state of water, its inversion is more difficult than with other approaches. Some recent studies have developed empirical models to investigate the direct correlation between TSI and reflectance [24] or some major water color indexes [3]. These models are simpler and more accurate, but their applicability and extrapolation performance require improvement. To overcome these difficulties, current research tends to use machine learning to develop TSI inversion algorithms [23].

However, most existing TSI inversion models have been developed based on lakes that are hyper eutrophic [23,25] and those in the medium trophic state have received insufficient attention. This skewed research focus may result in a biased assessment of the health of certain lakes because hyper eutrophic lakes have nutrient concentrations that significantly exceed the tolerance limit of the ecosystem and are accompanied by changes in water color and algal blooms [25]. In contrast, lakes in the medium trophic state have high nutrient levels but are within a controllable range and have shown an increasing trend toward eutrophication in recent decades [26]. Furthermore, large lakes have complex water color and hydrodynamic characteristics, and their nutrients are exchanged violently with water flows. They are rich in colored, dissolved, organic, and suspended matter and have complex optical properties. Therefore, it is difficult to construct a highly accurate TSI inversion model applicable to different trophic states using spectral information alone, and in our study we consider the use of environmental factors, such as pH, temperature (T), average wind speed (AWS), wind direction (WD), precipitation (P), DO, sediment content (SC), sediment transport rate (STR), etc., to compensate for this deficiency.

Based on the above background, our research objective was to develop a comprehensive integrated framework for the trophic state inversion of water bodies, as shown in Figure 1. Specifically, we focused on (i) combining regional remote sensing data and the key environmental factors to compensate for the uncertainty of the traditional remote sensing inversion model, (ii) developing a TSI inversion model applicable to large lakes in different

trophic states, and (iii) using relevant optimization algorithms to improve the accuracy of the model and applicability of the framework. We used Hongze Lake as a case study to provide a practical example of how this integrated framework could be used. The aim was to achieve the quantitative restoration of lake ecosystems.

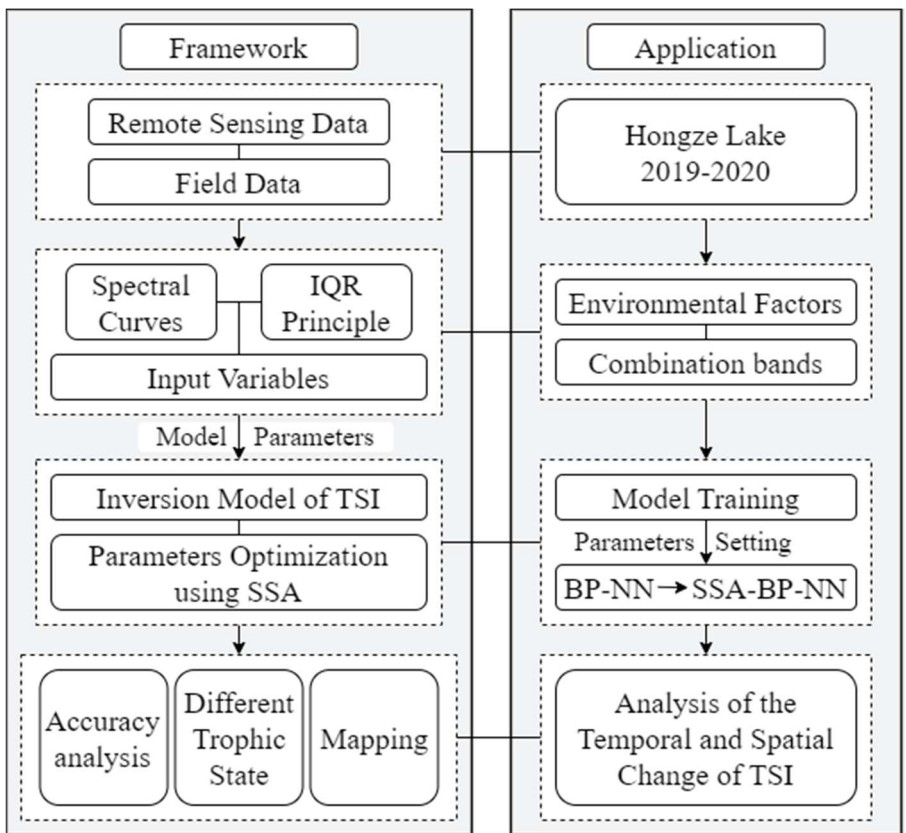

**Figure 1.** Integration framework design. IQR refers to the interquartile range principle, and SSA refers to the sparrow search algorithm.

## 2. Study Area and Datasets

### 2.1. Study Area

Hongze Lake (33°03′–33°40′N and 118°10′–118°52′E), one of the largest freshwater lakes in the lower reaches of the Huaihe River, is located in the western area of Jiangsu Province, China, and has a surface area of approximately 1580 km$^2$ [27]. The bottom of Hongze Lake is flat and consists mainly of sediment and phytoplankton. As an important source of the Huaihe River and an important reservoir for the South–North Water Transfer Project, Hongze Lake plays an important role in climate regulation, ecological protection, flood and drought prevention, agricultural irrigation, and as a source of drinking water [28].

The trophic state of Hongze Lake is mainly influenced by human activities such as industrial production, the use of fertilizers and pesticides, and urban sewage discharge. These activities lead to excessive accumulation of nutrients in the soil and water, which then flow into the lake and gradually worsen eutrophication in Hongze Lake.

Although the field monitoring data show that the nutrient concentrations in Hongze Lake, such as TP and TN, are slightly decreasing year by year [29], they remain at a high level. The problem of eutrophication cannot be ignored, and there is a risk of algal blooms which may harm the lake ecosystem and endanger human safety and production activity [25]. Therefore, we took Hongze Lake as an example and used the integrated framework that we had developed to assess its trophic state under the influence of complex environments.

## 2.2. Data Collection

The period of interest for this study is 2019–2020 when river dredging and remediation projects were implemented in the Hongze Lake basin to remove redundant sediment from the rivers entering the lake. A series of engineering measures contributed to improving the mobility of water bodies, thereby improving the trophic state level.

### 2.2.1. Field Measurements

To comprehensively and objectively assess the environmental characteristics and trophic state of the water in the study region, 16 sampling stations were distributed evenly across the lake (Figure 2). From January 2019 to December 2020, we conducted a total of 24 in situ field monitoring exercises to obtain hydrological and water quality data in the middle and latter half of each month. Water environment indicators such as pH, DO, SC, TP, TN, Chl-a, $COD_{Mn}$, and SD were obtained, and national water quality monitoring requirements were strictly adhered to during the sampling, monitoring, and laboratory analysis processes for all indicators.

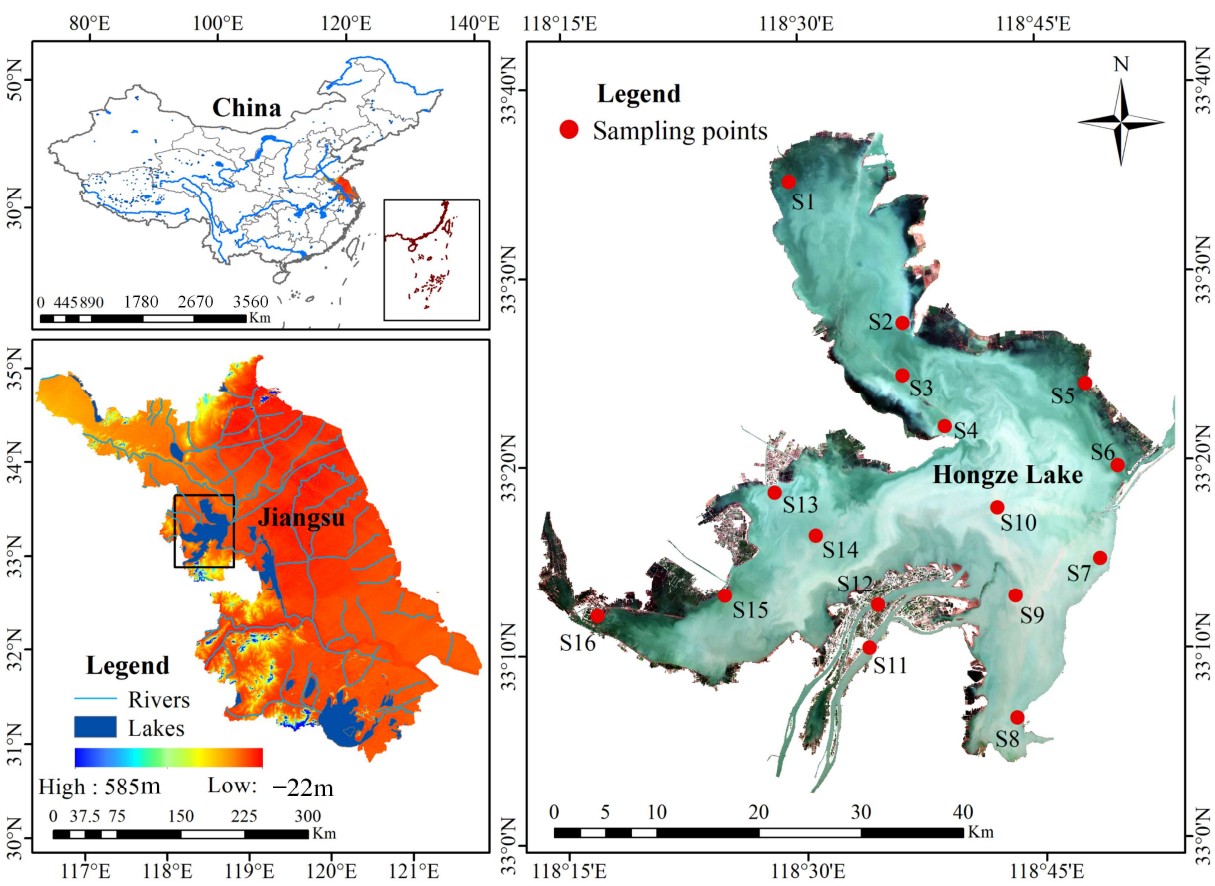

**Figure 2.** Map of Hongze Lake and distribution of monitoring stations.

### 2.2.2. Remote Sensing Data

Sentinel-2 (S-2) is a high-resolution, multispectral imaging satellite consisting of two complementary satellites, S-2A and S-2B, which were launched in 2015 and 2017, respectively, together with a multispectral imager (MSI) sensor which contains 13 bands. The European Space Agency (ESA) offers free downloads of satellite images. Compared to other satellites such as MODIS, MERIS, and Landsat-8/OLI [30], S-2/MSI has a higher spatial resolution (10 m, 20 m, and 60 m), and its revisit period is 5 days. The feasibility of the TSI inversion algorithm varies considerably between different sensors because of their various band configurations [31]. However, S-2/MSI has 3 special spectral bands (vegetation red edge) that can be used to monitor trophic state changes in lakes [18,32].

For trophic state assessment, this study retrieved 24 S-2/MSI images from the ESA Data Center between January 2019 and December 2020. Because the study area was often largely obscured by clouds, a total of 22 cloud-free images covering the study area were obtained and used to construct the TSI inversion model. Moreover, the time interval between these images and field monitoring data was less than five days.

### 2.2.3. Meteorological Data

Meteorological data corresponding to the sampling time were obtained from the Hydrological Yearbook of the People's Republic of China and NOAA's National Climatic Data Center. Variables of interest included T, P, WD, and AWS.

## 3. Methods

### 3.1. An Integrated Framework for TSI Inversion

An integrated framework employs multiple techniques and methods to invert the TSI of water bodies. Based on the framework, the specific workflow for this study was as follows (Figure 3):

1. Step 1: Data collection and collation, and identification of the main driving factors of TSI. Complex environmental factors were used to compensate for the uncertainty of remote sensing inversion.
2. Step 2: Quantification of the degree of influence of driving factors on TSI. Key environmental factors were selected as the optional input variables for the model.
3. Step 3: Data preprocessing, outlier cleaning, etc., to obtain the data set.
4. Step 4: Construction and optimization of the TSI inversion model.
5. Step 5: Model accuracy assessment and temporal and spatial distribution mapping.

### 3.2. Data Preprocessing

### 3.2.1. Region of Interest Extraction

In this study, we used the B4, B3, and B2 of S-2/MSI images for true-color synthesis and the support vector machine (SVM) method, a classification algorithm, was used to extract water bodies (Figure 2). The SVM can achieve a balance between training accuracy and promotion performance, effectively avoid the subjectivity of human-set thresholds (e.g., NDWI [33], MNDWI [34], and AWEI [35]), and is more suitable for water bodies with complex lake boundary structures and high feature space variability [36].

### 3.2.2. Preprocessing of Remote Sensing Images

In this study, preprocessing steps such as radiometric calibration and atmospheric correction of S-2/MSI images were performed using the Sen2cor toolbox [23] to obtain images of level L2A, which were used to characterize the surface water reflectance in the study area. Additionally, SNAP and ENVI 5.6 were utilized for image preprocessing [15], which included resampling to 10 m and image cropping, and due to the large area of Hongze Lake and the MSI operational orbit settings, the S-2 single image could not cover the study area. Therefore, it is necessary to use the Mosaicking toolbox in SNAP to mosaic the two images from the same date. In addition, different spectral bands have different sensitivities to the reflection and absorption properties of water bodies. To improve the accuracy of the model, we processed band combinations of S-2/MSI data to improve the response relationship between S-2/MSI data and TSI using the complementarity of scientific principles between bands [13]. According to the correlation coefficients, 12 feature combination bands were selected as the input variables of the model, and to avoid a high degree of co-linearity between the input variables they were not exactly used the same bands in the calculation process. The absolute values of the correlation coefficients for the selected feature combination bands were all in the range of 0.29–0.38 (Table 1).

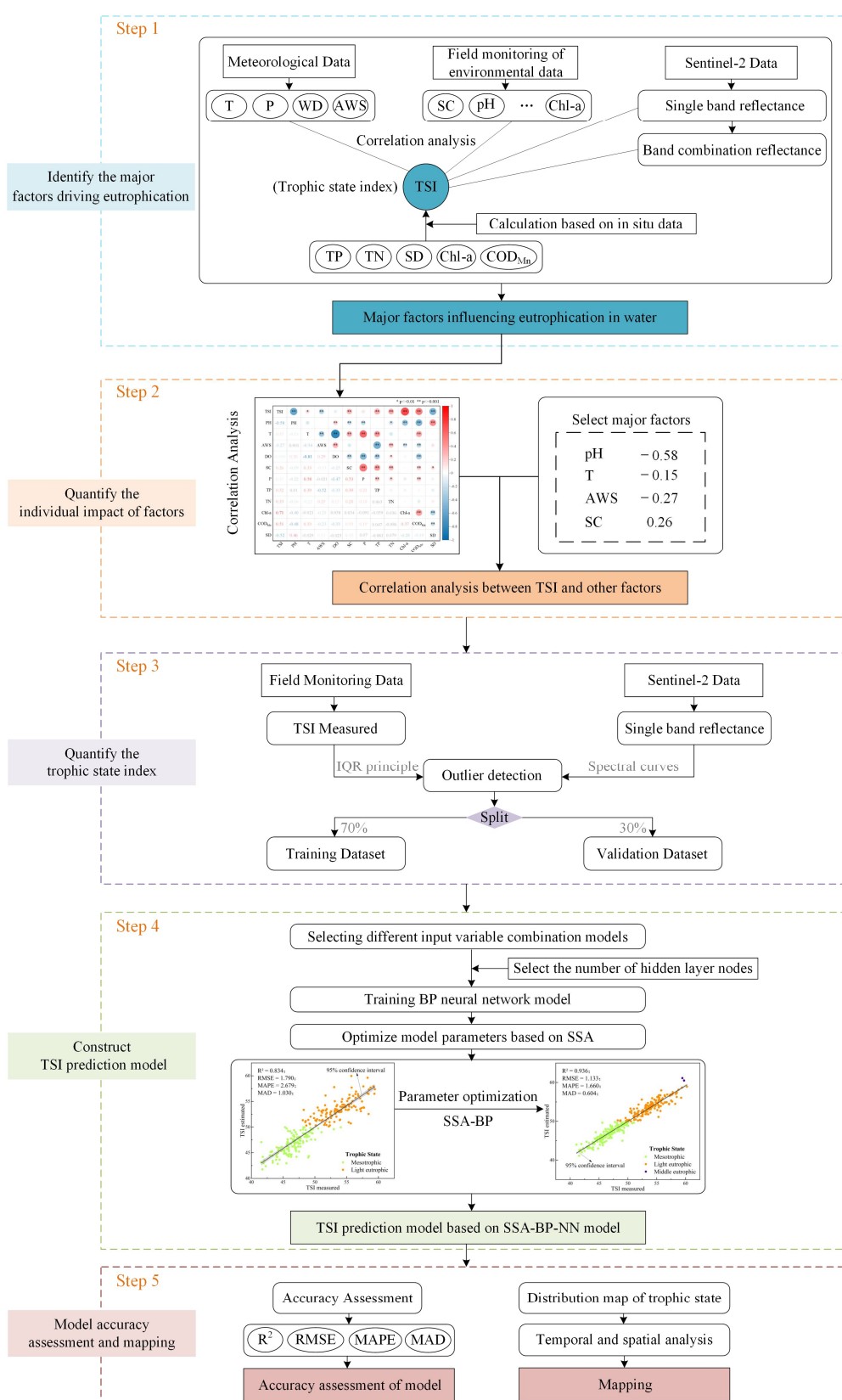

**Figure 3.** Flowchart showing the application of the framework in this study.

**Table 1.** Correlation between band combination of S-2/MSI images and TSI.

| No. | Band Combination | Correlation | No. | Band Combination | Correlation |
|---|---|---|---|---|---|
| 1 | (B11 − B8)/(B11 + B8) | −0.317 * | 7 | B12/B11 | 0.311 * |
| 2 | (B8A − B8)/(B8A + B8) | −0.369 ** | 8 | (B8A − B6)/(B8A + B6) | −0.352 ** |
| 3 | (1.5B11 − (B8 + B3)/2)/(1.5B11 + (B8 + B3)/2) | −0.308 * | 9 | (B8A − B7)/(B8A + B7) | −0.380 ** |
| 4 | B4/B1 | 0.305 | 10 | B9/B3 | −0.305 |
| 5 | B8A/B5 | −0.291 | 11 | B9/B4 | −0.306 |
| 6 | B9/B5 | −0.335 * | 12 | (B9 + B11)/(B3 + B4) | −0.300 |

Pearson's correlation coefficient. When a correlation is greater than 0, it means that the band combination is positively correlated with the TSI value, and a correlation less than 0 indicates a negative correlation. The greater the absolute value of correlation, the higher the correlation between the TSI value and the band combination. * and ** indicate a significant correlation at the 0.05 (double-tailed), and 0.01 (double-tailed) levels, respectively.

### 3.2.3. Spectral Curve Outlier Removal

Large lakes have significant temporal and spatial variability in the distribution of TSI and optically active substances in the water bodies because of variations in their genesis and environment. Therefore, the optical characteristics of various regions vary considerably [37]. The spectral curve is the energy distribution of light absorbed or emitted by water, which is mainly determined by the components and optical characteristics of the water. Additionally, when the water is in different trophic states, the reflectance of different bands has different properties of absorption, scattering, and transmission [38]. To analyze its optical characteristics, reflectance data extracted from S-2/MSI images were plotted as spectral curves.

As shown in Figure 4a, there were nine monitoring points with spectral curve trends which differed from the other points. The corresponding remote sensing pixel points revealed that the degree of cloud coverage had a significant impact on these points, with a cloud confidence degree of more than 25. Additionally, water vapor and aerosols had an impact on the band reflectance. There were certain errors in the preprocessing of the S-2/MSI images, and there were floating objects such as leaves in the lake, leading to errors in the spectral curves at some points that were difficult to avoid. Therefore, the nine cloudy monitoring points were eliminated.

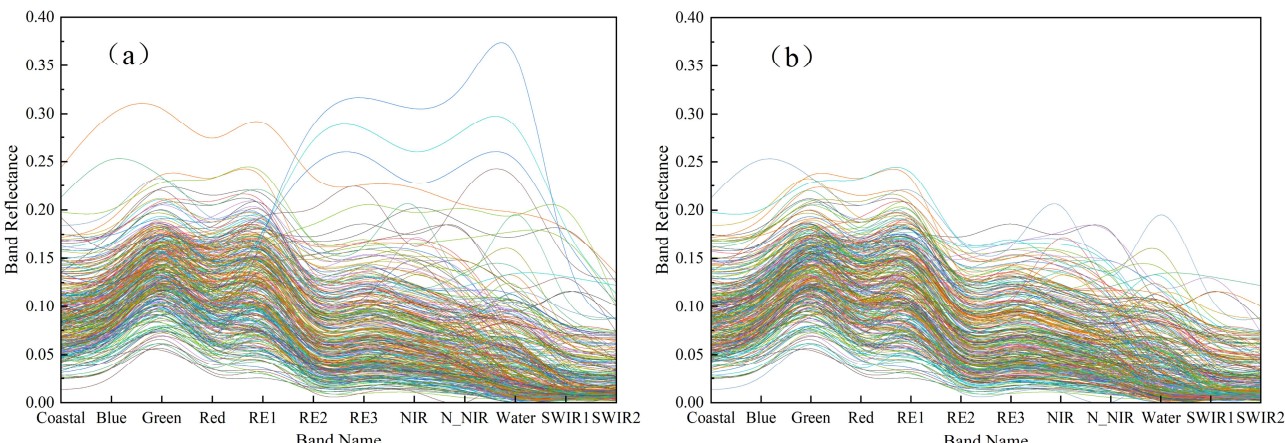

**Figure 4.** (**a**) The original spectrum curve and (**b**) the spectrum curve after the abnormal value is filtered.

Figure 4b shows that the remaining monitoring points had the same spectral curve trend. Additionally, the overall spectral reflectance of the water in Hongze Lake was between 0 and 0.3, and the spectral curve reached the reflectance peak near B3 (Green) and B5 (vegetation red edge) before gradually decreasing and approaching 0.

### 3.2.4. TSI Outlier Removal

Because of unavoidable mistakes in monitoring, transmission, or transcription, field monitoring datasets usually contain outliers which deviate from the normal range [39]. To

ensure the reliability and accuracy of the datasets used in machine learning, it is necessary to remove outliers from the large amount of data used in the study.

The interquartile range (IQR) principle of the box plot utilizes a nonparametric statistic to identify outliers based on quartiles and interquartile ranges. This statistic has no restrictions on the original data set, applies to all distribution types of outlier measurement, and provides a more objective way to identify outliers. Therefore, in this study, we adopted the IQR rule to clean the water environment data set.

The IQR rule sorts the original data from lowest to highest, taking 1/4 of the positions as the lower quartile (Q1), 1/2 of the positions as the median (Q2), 3/4 of the positions as the upper quartile (Q3), and the IQR as the difference between Q3 and Q1. When the original value is greater than the sum of Q3 and 1.5 IQR or less than the difference between Q1 and 1.5 IQR, it is considered an outlier.

As shown in Figure 5, there are two outliers in the TSI values calculated using the five field monitoring water quality indicators, which occurred in April 2019 (TSI = 48.48) and August 2019 (TSI = 65.96) and at sampling points S5 and S8, both of these are close to the shore and more affected by human activities.

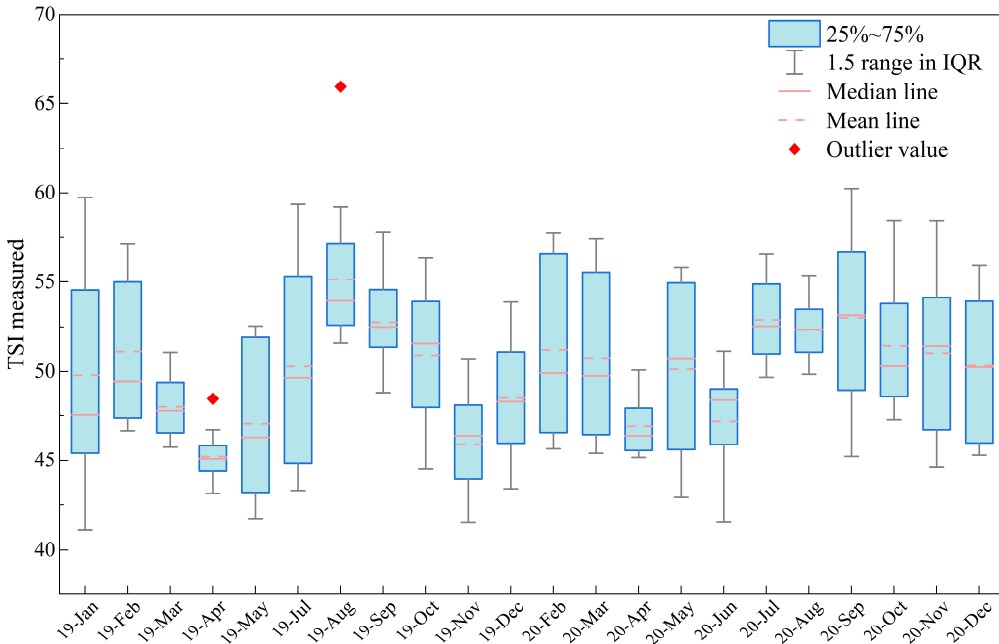

**Figure 5.** The monthly TSI distribution was calculated using in situ sample data monitoring five key indicators between January 2019 and December 2020. Each box shows the TSI between Q1 and Q3. In the boxes, the pink solid line indicates the median of TSI, the pink dotted line indicates the mean of TSI, the endpoints indicate the 1.5 IQR, and the red point indicates an abnormal value.

Overall, the TSI of Hongze Lake was higher in summer than other seasons. In addition to seasonal differences, TSI showed spatial differences. During the study period, the Hongze Lake sampling point S5 was in light eutrophic state, with TSI of approximately 52.5. The water body of Hongze Lake represented by S11 and S12 sampling points were connected to the Huaihe River and influenced by its change of the trophic state and sediment content; as a result, the trophic state of these two points were higher than that of other points, and their TSIs were 53.7 and 54.2, respectively.

### 3.3. TSI Inversion Model Based on Backpropagation Neural Network

Because of the complexity of the optical properties of the water in large lakes, the relationships between environmental data, meteorological data, S-2/MSI data, and TSI are complex and nonlinear. The BP-NN has significant advantages in solving nonlinear problems [40].

In this study, we initially constructed the TSI inversion model based on BP-NN, which includes an input layer, implicit layer, and output layer. The input layer consists of highly correlated meteorological and environmental factors as well as 12 feature combination bands (RS) based on S-2/MSI data, and the output layer is the TSI inversion values. To improve the generalization ability of the model, accelerate its training speed, and alleviate the problem of gradient disappearance or explosion for this study, we normalized the data in the input and output layers. The data normalization equation is as follows:

$$y = \frac{x - x_{Min}}{x_{Max} - x_{Min}} \tag{1}$$

where $x$ is the measured value of the corresponding data for the input and output layer parameters, $x_{Min}$ and $x_{Max}$ are the maximum and minimum values of the sample data, and y is the normalized value corresponding to $x$.

The training number of the BP-NN model was set to 10,000 times, the learning rate was set to 0.001, and the remaining parameters were kept as default values. To achieve the best model-fitting effect, the model also tested the number of hidden layer sizes.

$$\text{HiddenLayerSize} = \sqrt{n + m} + \alpha \tag{2}$$

where $n$ and $m$ are the numbers of neuron nodes in the input and output layers, respectively, and $\alpha$ is a random number between 1 and 10.

### 3.4. Optimization of Model Parameters Based on Sparrow Search Algorithm

The sparrow search algorithm (SSA) is a new swarm intelligence optimization algorithm proposed in 2020 [41], based on the behavior of sparrows foraging and evading predators, which has fewer control parameters, higher convergence performance, and local search capability. The SSA algorithm outperforms the gray wolf optimization (GWO), gravitational search algorithm (GSA), and particle swarm optimization algorithm (PSO) in terms of accuracy, convergence speed, and stability [42,43].

Therefore, in this study, SSA was chosen to optimize the initial BP-NN weights and thresholds. The algorithm divides the population into producers and followers, and the producers in the model were set to 20% of the population. The number of sparrows was 5, the safety threshold (*ST*) was 0.8, and the maximum number of iterations was 50. Producers typically have high energy reserves and are responsible for searching for areas with abundant food, providing foraging direction for all followers. When the alert value $R_2$ is greater than *ST*, the sparrows engage in antipredator behavior and update the position of the population to find an optimal solution.

The location of the producers was updated as follows:

$$x_{i,j}^{t+1} = \begin{cases} x_{i,j}^t \times \exp(\frac{-i}{\alpha \times iter_{\max}}) & R_2 < ST \\ x_{i,j}^t + Q \times L & R_2 \geq ST \end{cases} \tag{3}$$

where $t$ is the current iteration, $x_{i,j}^{t+1}$ is the position of the *i*-th generation sparrow in the *j*-th dimension in the *t*-th iteration, $\alpha \in (0,1]$ is a random number, $iter_{\max}$ is the largest number of iterations, $R_2 \in [0,1]$ and $ST \in [0.5,1.0]$ are random numbers, $Q$ is a random number obeying normal distribution, $L$ is an all-1 matrix of $1 \times$ dim, and dim refers to the matrix dimension. When $R_2 < ST$, it means that there is no predator within the foraging range and producers can search for food extensively; when $R_2 \geq ST$, it means that predators appear and all producers need to fly to a safe area.

The location of the followers was updated as follows:

$$x_{i,j}^{t+1} = \begin{cases} Q \times \exp(\frac{x_{wj}^t - x_{i,j}^t}{i^2}) & i > \frac{n}{2} \\ x_p^{t+1} + \left| x_{i,j}^t - x_p^{t+1} \right| \times A^+ \times L & i \leq \frac{n}{2} \end{cases} \tag{4}$$

where $x^t_{wj}$ is the location of the sparrow with the worst adaptation in the *t*-th iteration and $x^{t+1}_p$ is the location of the sparrow with the best adaptation in the *t* + 1-th iteration. A denotes a matrix of 1 × dim with elements defined randomly as 1 or −1, and $A^+ = A^T(AA^T)^{-1}$. When $i > n/2$, it indicates that the *i*-th follower has low fitness and is not eligible to compete for food with the producer, and when $i \leq n/2$, the follower will find food at the optimal individual.

Based on the anti-predatory behavior of sparrows, the following formula is used to update the location of the sparrow population:

$$x^{t+1}_{i,j} = \begin{cases} x^t_{bj} + \beta \times \left| x^t_{i,j} - x^t_{bj} \right| & f_i \neq f_g \\ x^t_{bj} + k \times \left( \frac{x^t_{i,j} - x^t_{bj}}{|f_i - f_w| + \varepsilon} \right) & f_i = f_g \end{cases} \tag{5}$$

where $x^t_{bj}$ is the global optimal position in the *t*-th iteration, $\beta$ is the parameter that controls the step size and obeys a normal distribution with a mean of 0 and a variance of 1. *K*, and $\in [-1, 1]$ is a random number. $f_i$ indicates the fitness value of the current individual, and $f_g$ and $f_w$ indicate the fitness values of the global best and worst individuals, respectively. When $f_i \neq f_g$, it means that the individual is at the edge of the population, and when $f_i = f_w$, it means that the individual is at the center of the population and needs to move closer to other individuals to stay away from danger.

### 3.5. Model Accuracy Assessment

The accuracy of the model can be assessed using the coefficient of determination ($R^2$), root mean square error (RMSE), mean absolute percentage error (MAPE), and mean absolute deviation (MAD). These four assessment parameters are calculated using the following formulae:

$$R^2 = 1 - \frac{\sum_{i=1}^n (t_i - y_i)^2}{\sum_{i=1}^n (t_i - \bar{t})^2} \tag{6}$$

$$RMSE = \sqrt{\frac{1}{n} \sum_{i=1}^n (t_i - y_i)^2} \tag{7}$$

$$MAPE = \frac{100}{n} \times \sum_{i=1}^n \left| \frac{t_i - y_i}{t_i} \right| \tag{8}$$

$$MAD = \frac{\sum_{i=1}^n |t_i - y_i|}{n} \tag{9}$$

where *t* is the TSI value calculated using the field monitoring data, *y* is the TSI value estimated with the model, *n* is the number of all samples, and *n* = 341.

### 4. Results

#### 4.1. Selection of Key Factors Driving Eutrophication

Based on several indicators collected in this study, we selected six typical meteorological and environmental factors for analysis: pH, T, AWS, DO, SC, and P. The Pearson correlation coefficients (PCC) between the TSI and these factors were calculated and the influence of these factors on the trophic state of the study area was assessed.

As shown in Figure 6, there was a significant correlation between pH and TSI, as well as between pH and key indicators (Chl-a, $COD_{Mn}$, and SD), with a PCC of at least 0.3. Using principal component analysis and redundancy, some researchers concluded that pH was the main environmental impact factor in Hongze Lake [44]. Therefore, pH (PCC = −0.58) can be used as an input variable. T was weakly correlated with TSI but T and other key indicators, such as TP and $COD_{Mn}$, had a strong correlation. Moreover, there was a correlation between TSI and AWS, and the PCC between AWS and TP was 0.52. According to previous research [45], T and AWS are the main factors contributing to eutrophication in water and the outbreak of algal blooms. Therefore, T (PCC = 0.15) and

AWS (PCC = −0.27) can be used as input variables for the model. In the study area, SC was significantly correlated with TSI, and there was a strong correlation between SC, TP, and TN. In addition, Hongze Lake is the product of the convergence of the Yellow River and the Huaihe River, and the volume of incoming sediment is greater than the outgoing sediment [46]. Hongze Lake shows an overall siltation trend. As a result, the dynamics of SC play a crucial role when studying the changing processes of the trophic state. SC (PCC = 0.26) was therefore used as one of the optional input variables in this study.

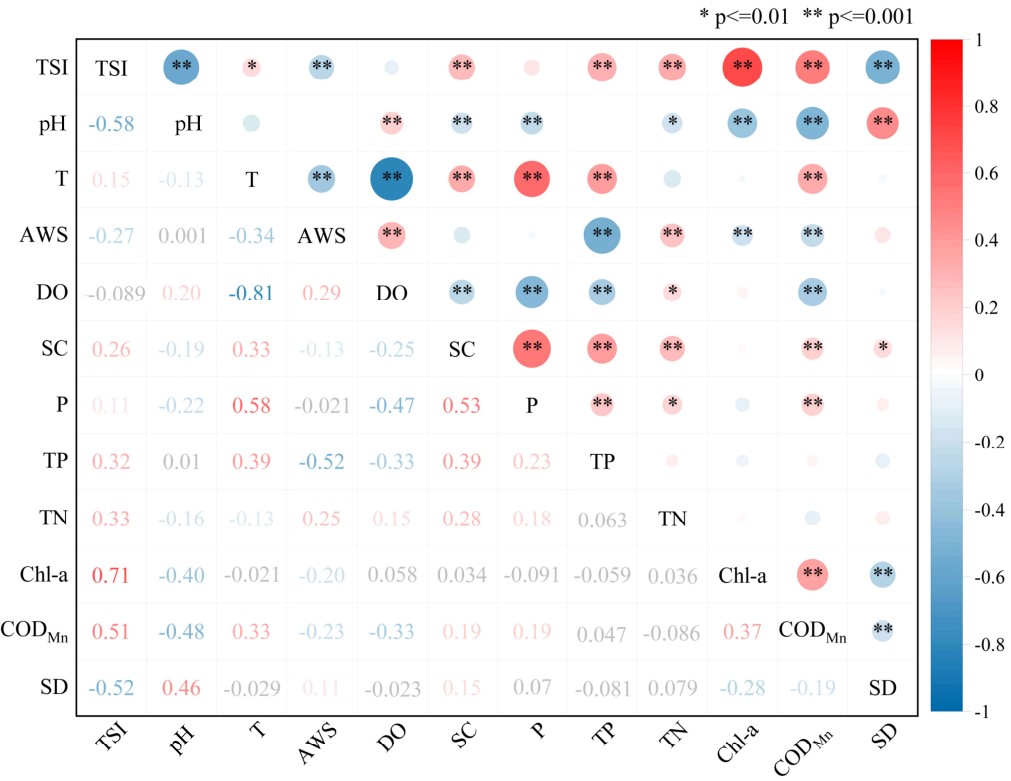

**Figure 6.** Correlation between TSI, field monitoring environmental indicators, and meteorological factors. For presentation purposes, correlation values below 0.1 are grayed out. * and ** indicate a significant correlation at the 0.01 (double-tailed) and 0.001 (double-tailed) levels, respectively.

Additionally, the PCC between DO and T was very high (PCC = −0.81). However, there was no significant correlation between DO and TSI, probably because the DO concentrations were primarily in the range of 8–12 mg/L with insignificant trends during the study period. Therefore, the impact of DO on TSI was excluded. The impact of P on TSI was disregarded because there was no significant correlation between P and TSI or other important indicators.

### 4.2. Trophic State of Water

A TSI-based method was proposed by the National Environmental Monitoring Center (NEMC) in 2001 [45] to assess the trophic state of rivers and lakes. This method integrates five trophic indicators: TP, TN, Chl-a, $COD_{Mn}$, and SD, where Chl-a is used as a reference parameter to determine the weighting coefficients of the five indicators in the TSI calculation, which are 0.1879, 0.1790, 0.2663, 0.1834 and 0.1834, respectively.

This study used TSI to quantify the trophic state of water bodies. The level of eutrophication increases as the TSI increases. The criteria for TSI classification are shown in Table 2:

**Table 2.** TSI classification standard.

| Trophic State Index Grading Range | Degree of Water Eutrophication |
|---|---|
| TSI < 30 | Oligotrophic |
| $30 \leq TSI \leq 50$ | Mesotrophic |
| $50 < TSI \leq 60$ | Light eutrophic |
| $60 < TSI \leq 70$ | Middle eutrophic |
| TSI > 70 | Hyper eutrophic |

According to the calculation results, the trophic state of Hongze Lake is more frequent in the mesotrophic and light eutrophic ranges, with a TSI between 40 and 60. Several monitoring sites were in the middle eutrophic (Figure 5).

### 4.3. Performance Comparison of the TSI Model with Environmental Factors

According to Figure 6, the four key factors available for the TSI inversion model were identified. These were randomly arranged and combined with RS to produce a total of 17 different combinations of input variables for the model. There were five groups; Zero-index indicates that S-2/MSI data only were used as input variables; Single-Index, Double-Index, Three-Index, and Four-Index indicate that n key factors and RS were combined as input variables, and n was 1, 2, 3, and 4, respectively. As shown in Table 3.

**Table 3.** Different input variable combination patterns considering environmental factors.

| Group | Input Variables |
|---|---|
| Zero-Index | No.1.Band reflectance, No.2.RS |
| Single-Index | No.3.pH&RS, No.4.T&RS, No.5.AWS&RS, No.6.SC&RS |
| Double-Index | No.7.pH&T&RS, No.8.pH&AWS&RS, No.9.pH&SC&RS, No.10.T&AWS&RS, No.11.T&SC&RS, No.12.AWS&SC&RS |
| Three-Index | No.13.pH&T&AWS&RS, No.14.pH&T&SC&RS, No.15.pH&AWS&SC&RS, No.16.T&AWS&SC&RS |
| Four-Index | No.17.pH&T&AWS&SC&RS |

Band reflectance refers to the single-band reflectance of S-2/MSI. RS refers to the reflectance of the 12 feature combinations shown in Table 1. No.2 can be used as a reference object to assess the accuracy-improving ability of other model input variables.

To optimize the accuracy of the model, avoid overfitting, and simplify the subsequent model training effort, this study employed No.17 as the basis for determining the hidden layer size of the model. As shown in Table 4.

**Table 4.** Influence of hidden layer size on accuracy of the model (No.17 pH&AWS&T&SC&RS). The No.7 test shows the optimal hidden layer size.

| No. | Hidden Layer Size | $R^2$ | RMSE | MAPE | MAD |
|---|---|---|---|---|---|
| 1 | 5 | 0.802 | 2.017 | 3.067 | 1.265 |
| 2 | 6 | 0.808 | 1.960 | 2.986 | 1.168 |
| 3 | 7 | 0.857 | 1.672 | 2.558 | 1.082 |
| 4 | 8 | 0.875 | 1.584 | 2.430 | 0.997 |
| 5 | 9 | 0.868 | 1.619 | 2.492 | 1.029 |
| 6 | 10 | 0.918 | 1.280 | 1.954 | 0.798 |
| **7** | **11** | **0.936** | **1.133** | **1.660** | **0.604** |
| 8 | 12 | 0.931 | 1.168 | 1.650 | 0.599 |
| 9 | 13 | 0.870 | 1.601 | 2.351 | 0.908 |
| 10 | 14 | 0.818 | 1.907 | 3.043 | 1.186 |

The results show that the model achieves its highest accuracy and the error tends to be stable when the hidden layer size is set to 11. Therefore, the hidden layer size in the subsequent model training was set to 11.

### 4.3.1. Performance Comparison of TSI Model Based on BP-NN

A total of 22 images were selected for the study after cloud coverage screening of S-2/MSI images. The S-2/MSI images were preprocessed, and a total of 352 samples were obtained covering the period from 2019 to 2020 by combining 16 monitoring locations in the study area (Figure 2). A total of 11 outliers were removed based on the spectral curve (Figure 4) and the IQR principle (Figure 5). Therefore, the model had 341 samples, of which 10% were randomly extracted as the test.

The distribution of the estimated and measured TSI when Zero-Index was used as the input to the BP-NN model is shown in Figure 7. It can be observed that the accuracy of the model improved to some degree after using the RS as input variables. The $R^2$ has improved from 0.322 to 0.469.

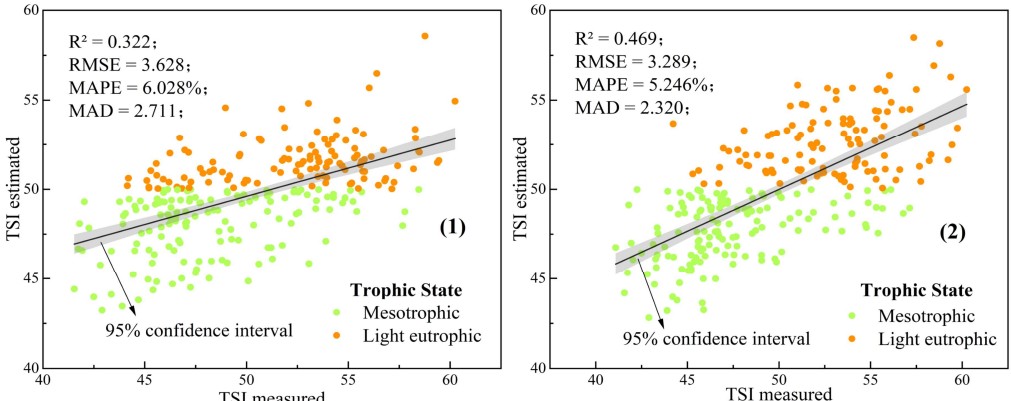

**Figure 7.** Performance of the TSI inversion model based on BP-NN (Zero-Index). The gray shaded area indicates the 95% confidence interval of the fit between the estimated and measured TSI. The number (1) and (2) correspond to the sequence numbers of the different input variable combination patterns in Table 3, and the serial numbers in Figure 8 through Figure 13 have the same meaning as in Figure 7.

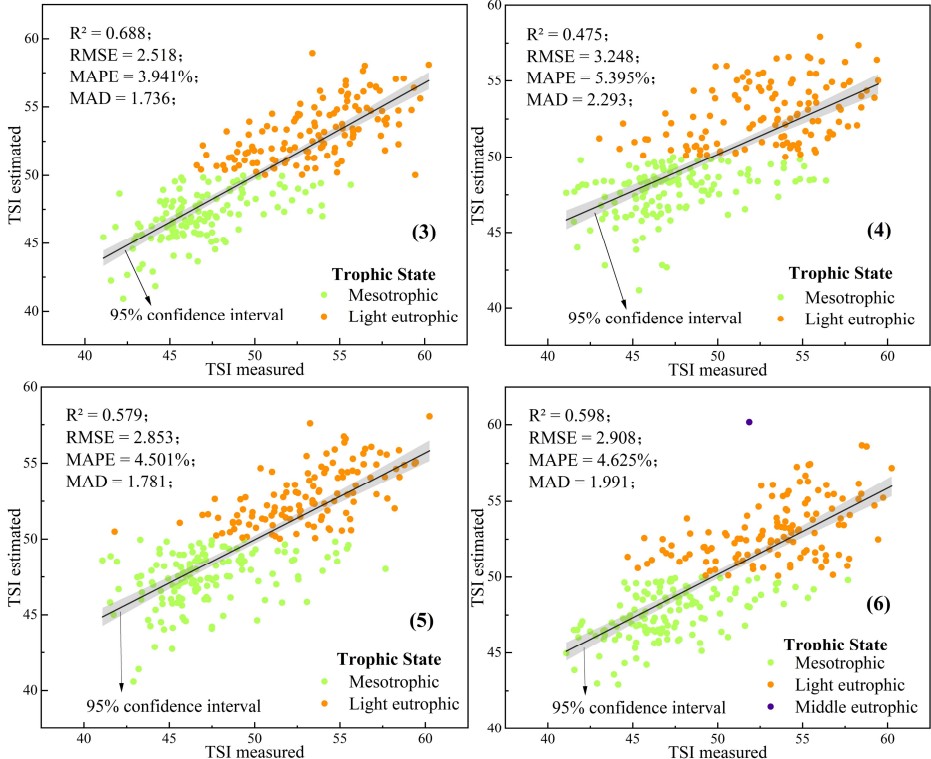

**Figure 8.** Performance of TSI inversion model based on BP-NN (Single-Index).

The results of Sinlge-Index being used as the input to the BP-NN model are shown in Figure 8. No.3 (pH&RS) had the highest accuracy ($R^2$ = 0.688, RMSE = 2.518, MAPE = 3.941%, MAD = 1.736), which was 46.7% better than reference object. No.4 (T&RS) had the lowest accuracy ($R^2$ = 0.475, RMSE = 3.248, MAPE = 5.395%, MAD = 2.293), which was only 1.3% better than the reference object. In addition, the accuracy of the model was similar when No.5 (AWS&RS) and No.6 (SC&RS) were used as the input variables.

The results of Double-Index being used as the input to the BP-NN model are shown in Figure 9. The accuracy ($R^2$) of the six different input variables ranged between 0.578 and 0.811. When pH was the dominant factor, the model was more accurate. No.8 (pH&AWS&RS) had the highest accuracy ($R^2$ = 0.811, RMSE = 1.939, MAPE = 2.836%, MAD = 0.996) which was 72.9% better than the reference object. No.11 (T&SC&RS) had the lowest accuracy ($R^2$ = 0.578, RMSE = 2.887, MAPE = 4.515%, MAD = 1.983) which was only 23.2% better than the reference object. In addition, it could be observed that when the Double-Index was used as the input variable, the accuracy of the model was greater than when the Single-Index was employed.

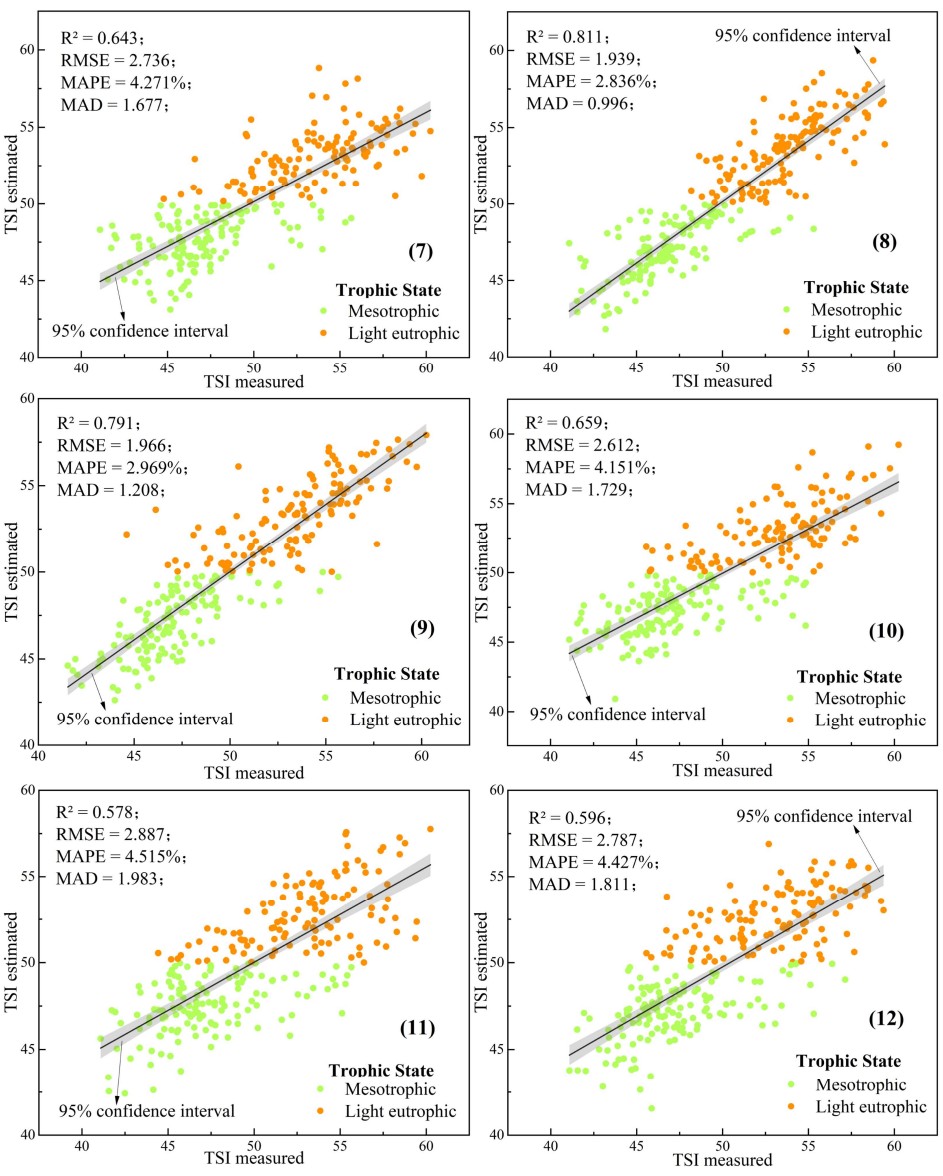

**Figure 9.** Performance of TSI inversion model based on BP-NN (Double-Index).

The results of Three-Index being used as the input to the BP-NN model are shown in Figure 10. The $R^2$ of the model was in the range of 0.66–0.81. No.13 (pH&T&AWS&RS) had

the highest accuracy ($R^2$ = 0.809, RMSE = 1.928, MAPE = 2.973%, MAD = 1.122), which was 72.5% better than the reference object. In addition, the accuracy of the model was higher when the Three-Index was used as the input variable than when the Double-Index was employed.

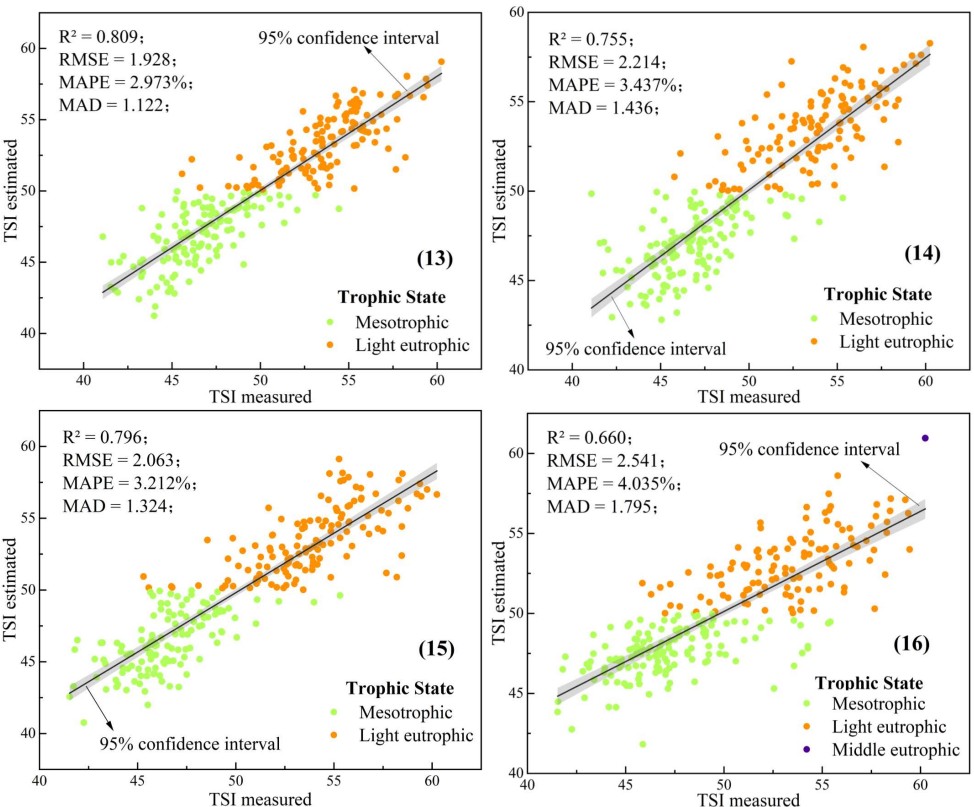

**Figure 10.** Performance of TSI inversion model based on BP-NN (Three-Index).

The results of Four-Index being used as the input to the BP-NN model are shown in Figure 11. This had the highest accuracy ($R^2$ = 0.834, RMSE = 1.790, MAPE = 2.679%, MAD = 1.030), which was 77.8% better than the reference object.

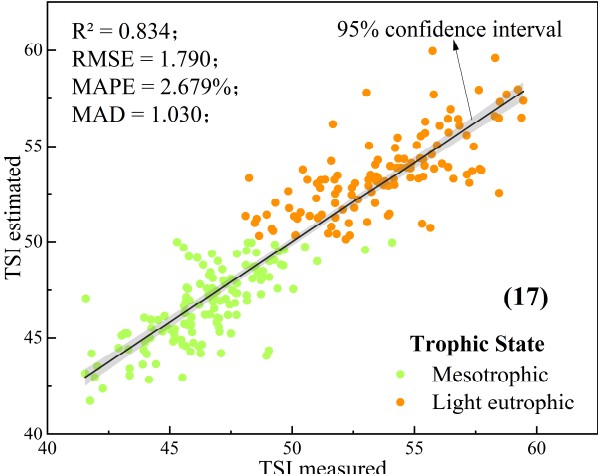

**Figure 11.** Performance of TSI inversion model based on BP-NN (Four-Index).

### 4.3.2. Performance Comparison of the TSI Model Based on SSA-BP-NN

When trained using Four-Index, the BP-NN achieved its highest accuracy of 0.834, which met the requirement but did not have outstanding advantages in the TSI inversion. SSA is used to optimize the weights and thresholds of the BP-NN model to improve its accuracy and generalization ability. Here, we trained the SSA-BP-NN model with Three-Index and Four-Index to explore the ability of SSA to improve the model.

The results of Three-Index being used as the input to the SSA-BP-NN model are shown in Figure 12. In particularly, the highest accuracy ($R^2 = 0.915$, RMSE = 1.325, MAPE = 1.964%, MAD = 0.739) was achieved when No.15 (pH&AWS&SC&RS) was used as input to the model, which met the accuracy requirements for TSI assessment. These results were significantly better than those of BP-NN models with the same input variables (Figure 10), for which the accuracy improved by 10.8%, 19.5%, 14.9%, and 17.9%, respectively.

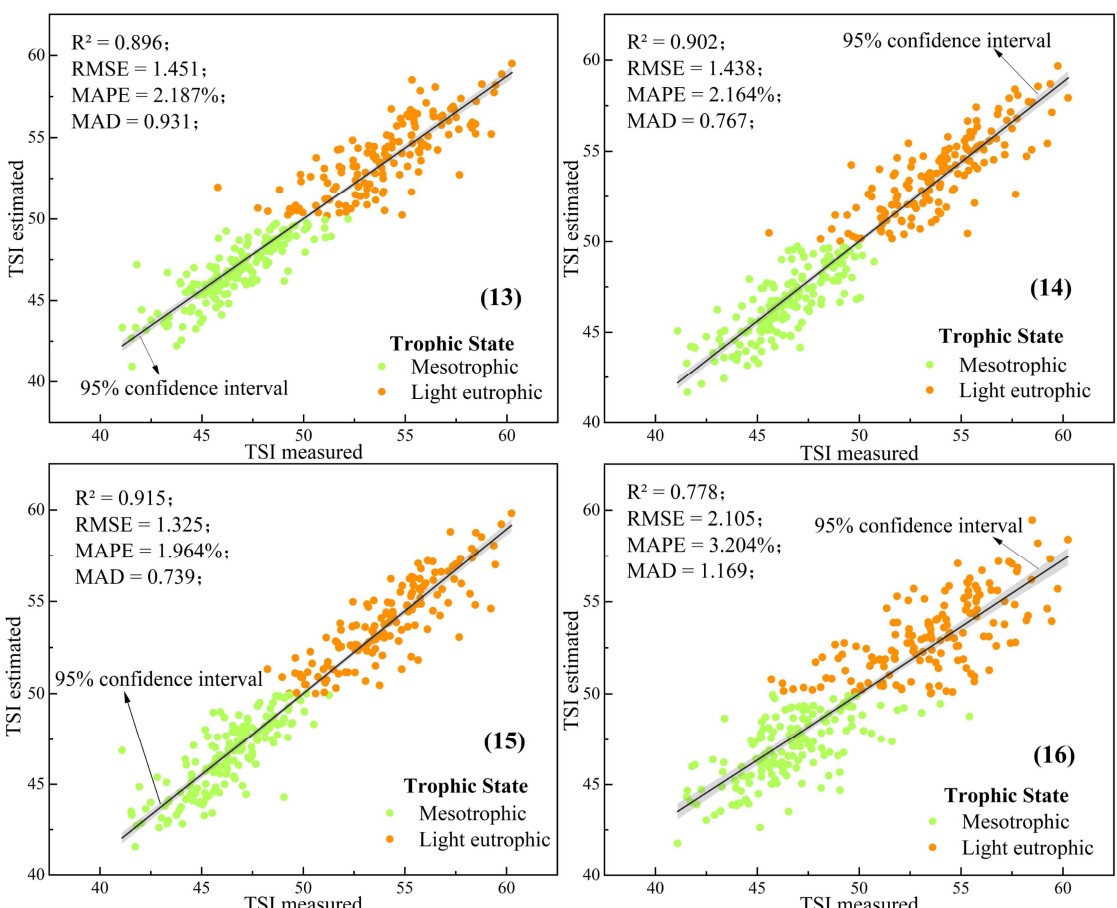

**Figure 12.** Performance of TSI inversion model based on SSA-BP-NN (Three-Index).

When Four-Index was used as the input, the SSA-BP-NN model achieved the highest accuracy ($R^2 = 0.936$, RMSE = 1.133, MAPE = 1.660%, MAD = 0.604), as shown in Figure 13. Compared to the BP-NN results, the overall accuracy increased by 12.3% (Figure 11).

In this study, 10% of the 341 samples were randomly selected as a test set to assess the generalization ability of the model. The results are shown in Figure 14.

In Figure 14, the TSI estimated with the SSA-BP-NN model and the TSI measured from the field data are indicated by the blue and red lines, respectively. It can be observed that the model fits well and can be applied to the TSI inversion and prediction of the trophic state in large lakes. It can provide a real-time early warning of the eutrophication level of the lake ecosystem if the environmental factors are guaranteed.

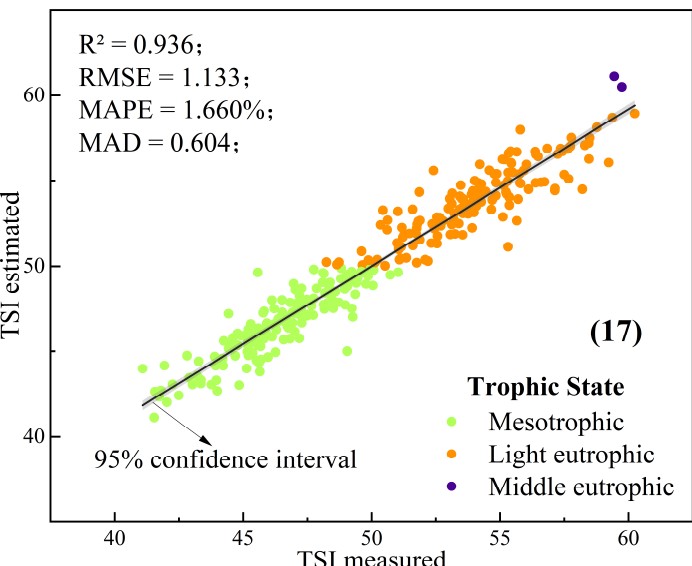

**Figure 13.** Performance of TSI inversion model based on SSA-BP-NN (Four-Index).

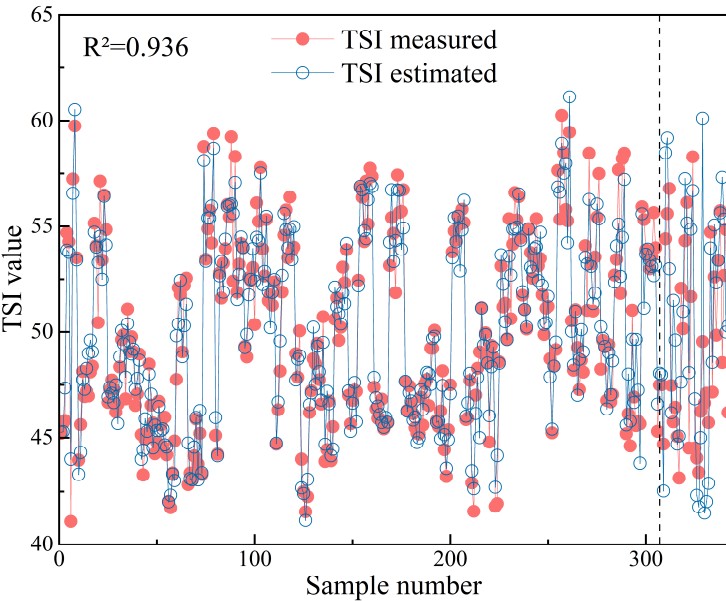

**Figure 14.** Model TSI prediction results and measured value distribution curve.

We calculated the accuracy of the model for different inputs as shown in Figure 15. The analysis results show that the TSI inversion model based on SSA-BP-NN established in this study has good accuracy enhancement capability and reliability, and the model error is within the acceptable range. In addition, the accuracy of the model improved by 6.7% and 79.5% for No.10 and No.1, respectively.

In addition, we further analyzed the inversion ability of the model for different trophic states of water when Three-Index and Four-Index were used as input variables. The results show (Figure 16) that the model identified the mesotrophic state ($30 \leq TSI < 50$) with the highest accuracy. Although the identification accuracy of light eutrophic ($50 < TSI \leq 60$) decreased slightly, the overall accuracy remained high. When Four-Index was used as the input to the model, the assessment errors of the different trophic states were small. When the eutrophication level of the water bodies was higher ($TSI > 60$), the assessment errors were smaller than the results when the Three-Index was used as input variables. This shows that the TSI inversion model established in this study is appropriate for the TSI inversion of large lakes in different trophic states.

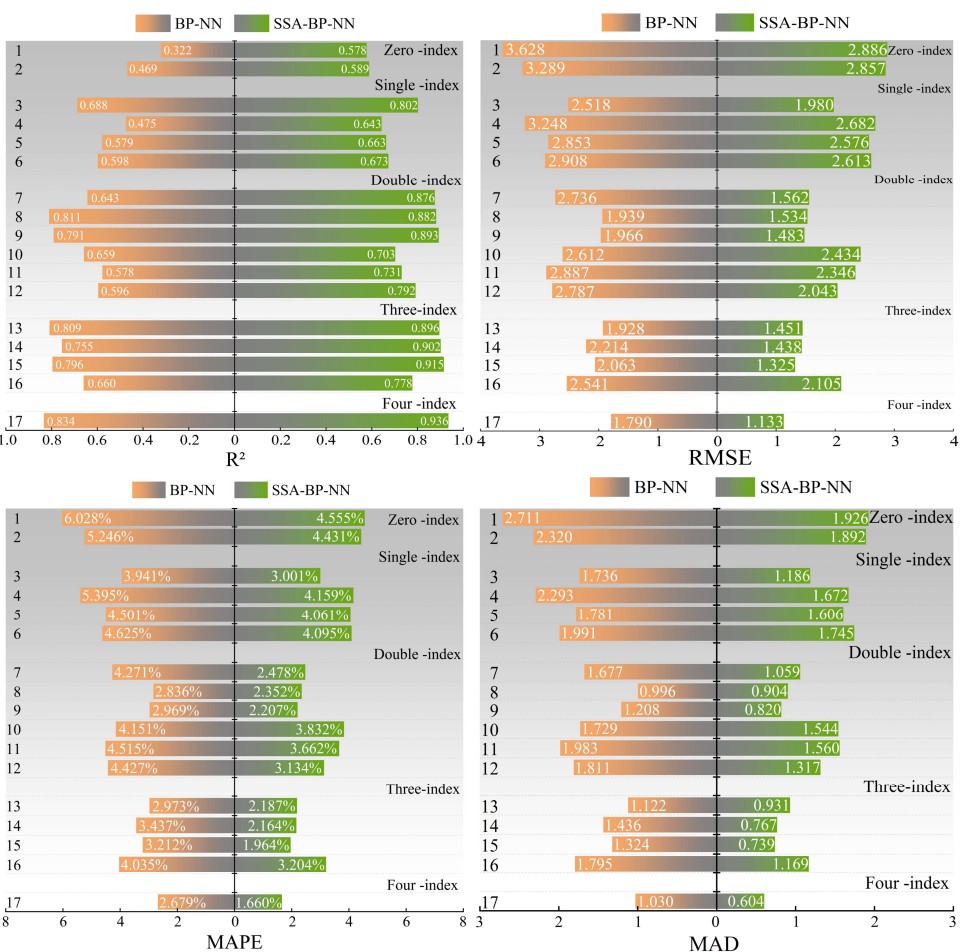

**Figure 15.** Accuracy comparison of different TSI inversion model inputs based on BP-NN and SSA-BP-NN.

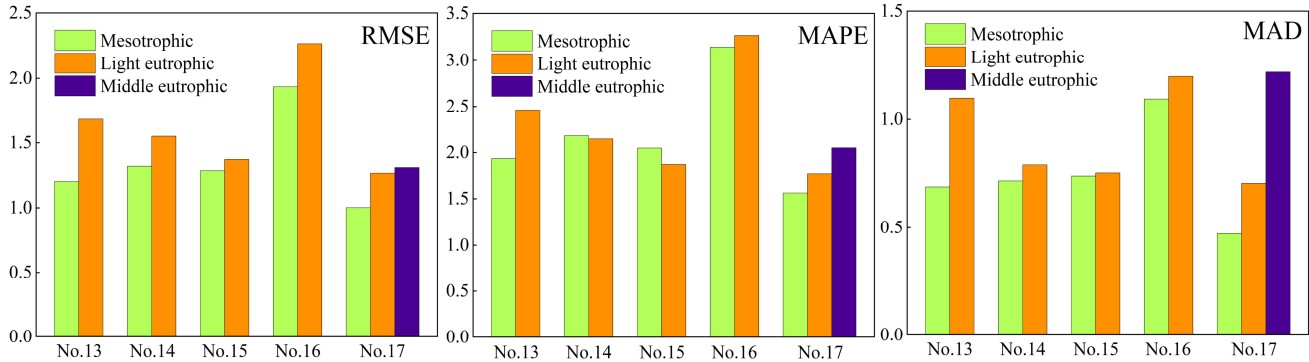

**Figure 16.** Accuracy of SSA-BP-NN model inversions for different trophic state levels. (Three-Index and Four-Index were used as the inputs to the model).

### 4.4. Temporal and Spatial Distribution of Trophic State

Here, using the comprehensive integrated framework, the TSI temporal and spatial distribution of Hongze Lake from 2019 to 2020 (Figure 17) was mapped by combining four key environmental factors and S-2/MSI data.

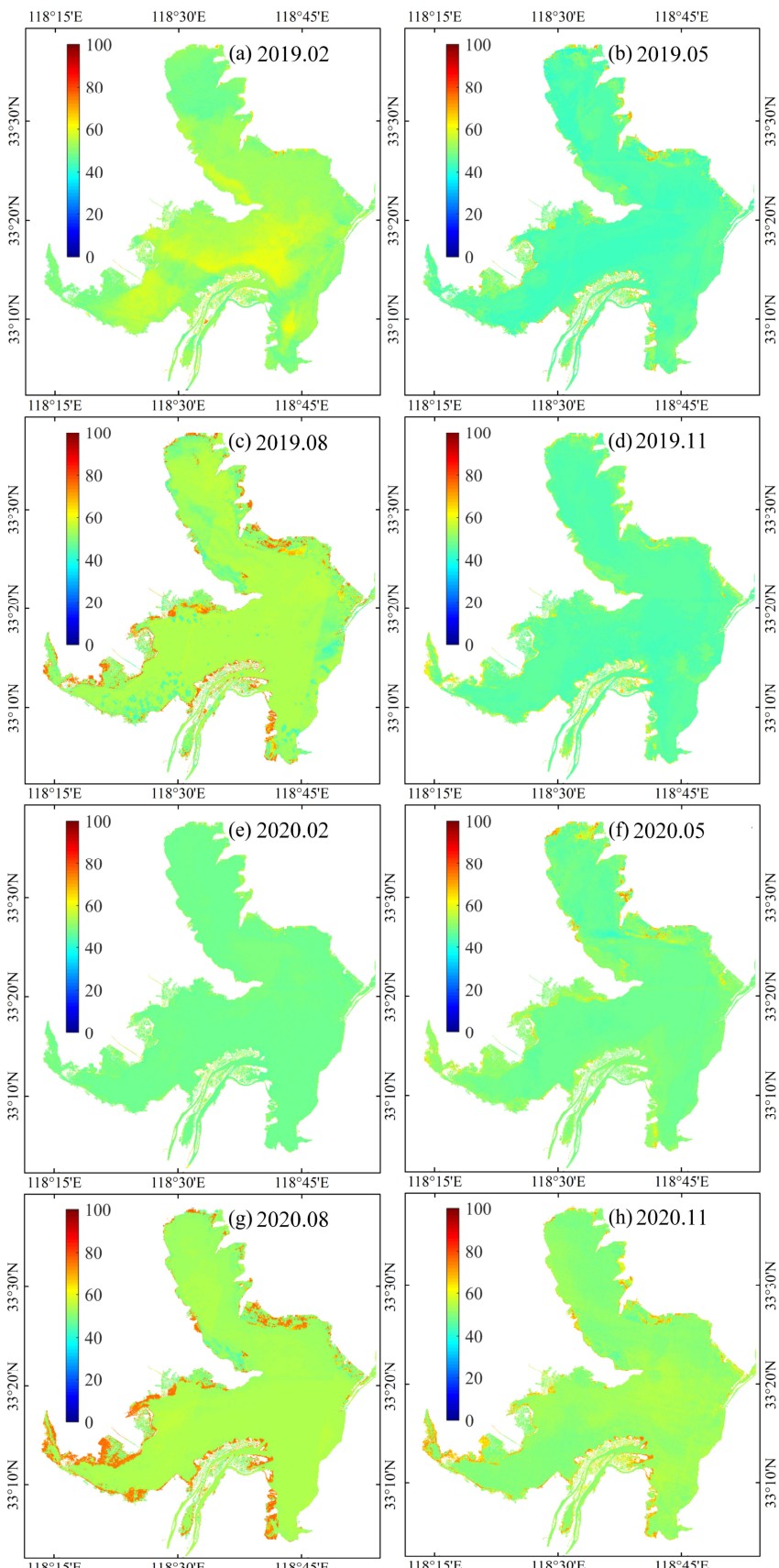

**Figure 17.** Temporal and spatial distribution of TSI in Hongze Lake from 2019 to 2020.

The model fitting results for the trophic state of Hongze Lake from 2019 to 2020 indicated that the whole lake had a rich trophic state during the summer and a poor trophic state during the winter. This is consistent with the changing trend of the water quality data. The primary reason for this phenomenon is the increase in temperature and rainfall within the study area during the summer. These factors lead to nutrients sinking into the lake via surface runoff, which restores the biochemical activity of the water and accelerates the growth rate, thereby intensifying the overall trophic state of the water.

It is worth noting that the trophic state of Hongze Lake in 2020 showed a decreasing trend compared with 2019, and this was most obvious in February. The results indicate that the river dredging and remediation project and the South–North Water Transfer Project conducted in the Hongze Lake basin were conducive to improving the trophic state of the study area.

Furthermore, the trophic state revealed more obvious spatial differences. The overall trophic state of the western and northern lake districts was higher than that of the eastern lake district [47]. The reason for this difference could be that the eastern lake district comprises the primary flowing water area of Hongze Lake, with higher water transparency, stronger water disturbance, and lower nutrient levels (e.g., TP, TN) than in the other areas, and therefore the degree of eutrophication is lower. The western and northern lake districts are relatively closed and less mobile. In addition, agriculture and fisheries are more developed around the western and northern lake districts and disturbance from human activities is greater, which makes nutrients such as algae multiply rapidly, resulting in a higher trophic state [48]. Moreover, with the water color change characteristics in Hongze Lake in Figure 2, the water color in the western and northern lake districts was substantially darker than that in the eastern lake district, confirming that the higher water mobility in the eastern lake and demonstrating that our results were accurate.

Unfortunately, the eutrophication level at the lake boundary of Hongze Lake is significantly higher than at its center, particularly in the summer. This may be due to the complexity of the lake edge and the difficulty in avoiding errors during the extraction of water, resulting in the misclassification of part of the land as a water body and the difference in optical properties between land and water. Equally, the water edge is more susceptible to human activities and other factors. Numerous factors contribute to a certain degree of error at the edge of our model results. Therefore, further refinement of the model is required to address such problems.

## 5. Discussion

### 5.1. Construction and Assessment of the TSI Inversion Framework

The integrated framework constructed in this study has broad application potential, which provides an integrated approach to the assessment of the trophic state of lake ecosystems. The framework is superior to existing inversion methods because it takes the special environmental characteristics of the region and incorporates field monitoring data, remote sensing data, multiple data processing methods, and machine learning algorithms. It also uses a comprehensive TSI indicator [22] which can more accurately invert the temporal and spatial variation characteristics of the trophic state. Firstly, through statistical analysis, we correlated the multisource data with TSI to identify the input variables of the model. Secondly, we constructed a nonlinear black box model by combining machine learning with new swarm intelligence optimization algorithms [40,41] with good convergence performance and local search capability. The application of this integrated algorithm effectively improved the prediction and generalization capabilities of the model [42]. Finally, according to the results of the case study, unlike previous studies, this integrated framework combined multi-source data with multiple methods and models, which avoids the limited scope of trophic state's identification due to the limitation of the S-2/MSI spectral bands and other factors [23,49]. This framework can not only be applied to lakes with a high frequency of algal bloom outbreaks but can also achieve high accuracy in the inversion and assessment of TSI for water bodies in a medium trophic state. This result shows that

our integrated framework can be applied for the assessment of the trophic state of lake ecosystems taking account of the special environmental features of the region and can provide a reference for the development of effective management strategies to improve the water quality of Hongze Lake and other similar lakes.

### 5.2. Selection of the Input Variables

In the process of constructing the framework, to overcome the limitation of using single or multiple indicators, we selected TSI as a comprehensive indicator to assess the trophic state of lakes. TSI includes several trophic indicators that can quickly assess the trophic state of lakes [23]. For S-2/MSI data, we eliminated outliers by analyzing the trend of the spectral curve. To increase the correlation between S-2/MSI data and TSI, the band combination of S-2/MSI data was performed on the basis of avoiding covariance [13,38]. For the field monitoring data, the IQR principle [23] was selected for data set cleaning to ensure the reliability of the input data. The use of these methods avoids the influence of outliers in the data set on the estimated results. In addition, through statistical analysis, we added new information sources (pH, T, AWS, and SC) that are important environmental factors affecting trophic state changes in the study area to enhance the adaptability of the framework. According to related studies [44–46], these indicators are all major variables influencing events such as eutrophication and algal bloom outbreaks in lakes. The combination of S-2/MSI data enables rapid identification of pollution sources and analysis of the temporal and spatial characteristics of trophic states under the influence of complex environments [9]. Based on this, we were able to achieve a trophic state identification accuracy higher than 90%. However, the application performance of the framework was limited by field monitoring when environmental factors were absent as model inputs. We encourage other researchers to apply the framework to other lakes to further optimize our approach.

### 5.3. Limitations and Future Perspectives

Although the integrated framework has strong application prospects, it nevertheless depends on remote sensing data and field monitoring data, and therefore its application may be limited in remote areas or for lakes that lack field monitoring data. In future research, more remote sensing sources can be considered to explore the use of emerging technologies such as the use of unmanned aerial vehicles to obtain high-resolution water quality data, thus further extending the application of the framework. In this study, key environmental factors were added to enhance the adaptability of the framework; however, only a few factors were taken into account. Lake ecosystems are extremely complex and are also affected by hydrodynamic processes, climate change, human activities, and other factors. To further understand the changing trophic state of lakes, other significant environmental factors could be explored and incorporated into the framework. Finally, the integrated framework primarily relies on machine learning algorithms for model construction and optimization, and other machine learning models, such as convolutional neural network (CNN), random forest (RF), logistic regression (LR), XGBoost, etc., could be considered for incorporation in future studies to further improve the accuracy and adaptability of the framework.

In summary, although the integrated framework achieved significant results in trophic state assessment, it still has some limitations. The adaptability of the framework to different lakes could be improved by further expanding data sources, taking additional representative environmental factors into consideration, and integrating multiple models.

## 6. Conclusions

In this study, our aim was to provide a comprehensive integrated framework which incorporated multiple data sources and methods and inverted the TSI indicator to assess the trophic state of lakes. We applied the developed framework to a typically large and complex lake in China (Hongze Lake). Key meteorological and environmental factors (pH, T, AWS, and SC) affecting the TSI were selected using statistical analysis and combined with

S-2/MSI data as input variables for the model. All of these indicators are closely related to the degree of eutrophication and can provide new sources of information. We constructed the SSA-BP-NN model and used $R^2$, RMSE, MAPE, and MAD to assess the performance of the model in this framework. When Four-Index was used as the input variables to the model before and after the dredging and remediation project in Hongze Lake during the study period ($n = 341$), the $R^2$ of TSI inversion reached 0.936, which could compensate for the shortcomings of remote sensing TSI inversion. The results of the case study indicate that the integrated framework could successfully assess the temporal–spatial trophic state of large lakes. In the future, more data may be acquired to optimize the inversion model in the framework and combine various machine learning and deep learning algorithms to improve its application and generalization capability.

**Author Contributions:** Conceptualization, D.M., J.M. and S.Z.; methodology, D.M. and H.G.; software, D.M.; validation, D.M.; formal analysis, D.M.; investigation, D.M.; resources, J.M.; data curation, D.M. and J.M.; writing—original draft preparation, D.M.; writing—review and editing, J.M. and W.L.; visualization, D.M.; supervision, J.M. and W.L.; project administration, J.M.; funding acquisition, J.M. All authors have read and agreed to the published version of the manuscript.

**Funding:** This research was funded by the Key Science and Technology Special Projects of Jiangxi Province, grant number 20213AAG01012; the National Key Research and Development Program of China, grant number 2019YFE0109900; and the Research funding of China Three Gorges Corporation, grant number 202003251.

**Data Availability Statement:** The data presented in this study are available on request from the corresponding author. The data are not publicly available due to the fact that it is currently privileged information.

**Acknowledgments:** The authors would like to thank the anonymous reviewers for their constructive comments and suggestions which strengthened a lot this paper.

**Conflicts of Interest:** The authors declare no conflict of interest.

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
