# Peer review of "An Integrated Framework for Remote Sensing Assessment of the Trophic State of Large Lakes"

_remotesensing, doi:10.3390/rs15174238_

Round 1
Reviewer 1 Report
This manuscript shows a comprehensive integrated framework combining remote sensing and environmental factors to assess the trophic state of large lakes, a very important integrated water quality index. The work should be of interest to other researchers and it’s clear that the manuscript is relevant to remote sensing assessment of water quality. The authors dealt with this interesting matter by developing an appropriate and innovative framework. Environmental factors and S-2/MSI data were combined to study the deterioration of trophic state of lakes from a new perspective; and it is interesting that a machine learning algorithm and sparrow search algorithm were also combined to optimize the trophic state assessment. Moreover, the models used in the integrated framework have some implications for other researchers.
Generally, the manuscript is well written; however, there are still some missing information about methods and the data pre-processing procedure. I suggest the authors consider the following remarks to further improve the manuscript.
Specific comments:
(1) Line 20: Which key environmental factors selected through statistical analysis you are talking about? The abstract should be more explicit about the key points of the manuscript so that the readers know it.
(2): Lines 24-25: The manuscript mentioned that the framework is capable of accurately identifying water bodies in a medium and hyper eutrophic state, which is a bit ambiguous. Is there a limitation to the application of the framework?
(3): Lines 64-67: The manuscript mentions combining multiple indicators to estimate the trophic state of water bodies. Can you elaborate on how multiple indicators were combined?
(4): Figure 1: The section of model training of the framework’s application in figure 1 is too simple, and it is recommended to increase the description of the relevant training process.
(5): Give more explanation on the samples and data of the S-2/MSI images used in the research. How many sampling points were employed in the framework and how is the TSI distributed at different points?
(6): Line 177: The description of the method for the section of region of interest extraction is simple, so please modify it accordingly.
(7): Line 191: Is MAI in “band combinations of S-2/MAI data” a spelling error?
(8): Lines 204-205: The introduction of the symbols are written in a redundant way, please make the modification.
(9): It was mentioned that the SSA algorithm is used to optimize model parameters in the manuscript. It is suggested to add more detailed introduction of the SSA method.
(10): What data sources do the authors use to calculate the figure 6?
(11): Line 353: The description of five key water quality indicators is not clearly distinguishable from the key environmental factors in the manuscript, and the authors reflect on how they might be modified to distinguish between these?
(12): The temporal and spatial distribution of TSI in Hongze Lake is shown in Figure 17, and the causes of the spatial variation of TSI are discussed. Can it be combined with the water color variation to explain the causes at a deeper level?
Round 2
Reviewer 1 Report
Thank you for addressing my comments and providing detailed responses. I am pleased with your replies and modifications, and I have no further suggestions for revisions.